# Degradation of Dibutyl Phthalate Plasticizer in Water by High-Performance Iro$_2$-Ta$_2$O$_5$/Ti Electrocatalytic Electrode

**Jia-Ming Xu [1], Shu-Hsien Chou [2], Ying Zhang [1], Mohanraj Kumar [2] and Shan-Yi Shen [2,***

[1] School of Resources and Environment, Northeast Agricultural University, 600 Changjiang Road, Harbin 150030, China; neauxjm@163.com (J.-M.X.); zhangying_neau@163.com (Y.Z.)

[2] Department of Environmental Engineering and Management, Chaoyang University of Technology, 168 Jifeng E. Rd., Taichung 41349, Taiwan; a7581052@gmail.com (S.-H.C.); mohan1991mpt@gmail.com (M.K.)

* Correspondence: syshen@cyut.edu.tw; Tel.: +886-4-2332-3000 (ext. 4210)

**Abstract:** Dibutyl phthalate (DBP) in the presence of a wastewater system is harmful to the environment and interferes with the human's endocrine system. For wastewater treatment, DBP is very difficult to be decomposed by biotechniques and many catalytic processes have been developed. Among them, the electrocatalytic oxidation (EO) technique has been proven to possess high degradation efficiency of various organic compounds in wastewater. In this study, an electrocatalytic electrode of iridium-tantalum/titanium (IrO$_2$-Ta$_2$O$_5$/Ti) was employed as the anode and graphite as the cathode to decompose DBP substances in the water. According to experimental results, the high removal efficiency of DBP and total organic carbon (TOC) of 90% and 56%, respectively, could be obtained under a voltage gradient of 10 V/cm for 60 min. Compared with other photocatalysis degradation, the IrO$_2$-Ta$_2$O$_5$/Ti electrode could shorten about half the treatment time and electric power based on the same removal efficiency of DBP (i.e., photocatalysis requires 0.225~0.99 KWh). Results also indicated that the production of hydroxyl radical (•OH) in the electrocatalytic electrode played a key role for decomposing the DBP. Moreover, the pH and conductivity of water containing DBP were slightly changed and eventually remained in a stable state during the EO treatment. In addition, the removal efficiency of DBP could still remain about 90% after using the IrO$_2$-Ta$_2$O$_5$/Ti electrode three times and the surface structure of the IrO$_2$-Ta$_2$O$_5$/Ti electrode was stable.

**Keywords:** dibutyl phthalate; electrocatalytic oxidation; iridium-tantalum/titanium; hydroxyl radical

## 1. Introduction

Due to industrialization's rapid development, industrial pollutants increase rapidly and endanger environmental safe and human health. Phthalates have been widely used as plasticizers and added to the plastic manufacturing process, and it is a chemical substance that humans come into contact with every day. The main products include PVC products, construction material, children's toys, printing inks and coatings, food containers, and packaging papers. Among the phthalates, dibutyl phthalate (DBP) is one of the most commonly used substances; particularly, industrial solvents, adhesives, waxes, inks, medicines, and cosmetics were found. Unfortunately, because of the incredible demand and biological toxicity of DBP, it has been considered as one of the primary harmful pollutants in U.S. EPA and belongs as an endocrine-disrupting substance (EDS). Even at very low concentrations, it will affect the human body's damage of endocrine, immune, and neurological effects [1,2]. With the lower covalent interaction between DBP and conventional plastics, the DBP is quickly released from the plastic matrix into the environment and causes accumulation, harming various aquatic organisms in the marine and freshwater. In Taiwan, the health and welfare ministry stipulates the tolerable daily intake (TDI) per person for the five most common plasticizers (such as DEHP, DBP, DINP, BBP, DIDP). Among them, the DBP has the lowest limit (i.e., 0.01 mg/kg bw/day). The TDI value is an international human daily tolerance for pollutants, which is used to estimate product

contamination content risk assessment. Besides, the discharge standard of Taiwan's effluent must reach 0.4 mg/L for DBP. In view of this, DBP has a high hazard and the urgency of treatment. Accordingly, developing an innovative and efficient treatment technology for DBP wastewater is highly expected. In the past, DBP degradation methods commonly were used, such as physical and biological. The physical process uses ion exchange and membrane separation technology or activated carbon to adsorb pollutants. Wang et al. used the phoenix leaf activated carbon (PLAC) to adsorb DBP from aqueous solutions through the chemical activation phosphate [3]; The maximum adsorption efficiency of DBP was 97% under pH 13, treatment time of 2 h, and room temperature. In the other study, the ginkgo leaves-activated carbon (GLAC) was chemically activated with zinc chloride to study the adsorption of DBP in an aqueous solution in an intermittent experiment. The adsorption efficiency of DBP can reach 97.5% at pH 13, 2 h operated, and room temperature conditions [4]. Although the physical method can effectively adsorb the DBP, it must be performed in a highly alkaline environment (pH 13), and the adsorbed product still needs subsequent treatment. The biological method was degrading the DBP by domesticating specific microorganisms; the Wu et al. study showed that the bacterial strain of SASHJ can effectively decompose DBP [5]; the DBP removal efficiency can reach 80% under pH 8 and 190 h of reaction time. However, the biological methods have issues of the domesticating microorganisms, environment pH, and treatment time. Moreover, the complete DBP mineralization still needs other microorganisms to be achieved. Therefore, it is necessary and urgent to develop an innovative technique for degrading DBP [6]. Recently, advanced oxidation processes (AOPs) such as the photocatalysis technique have been used to degrade DBP. The catalyst materials are irradiated at specific ultraviolet wavelengths, and the hydroxyl radicals will be generated and further oxidize the DBP in the water. For instance, Fe(III) could degrade the DBP under 365 nm irradiation and reach a removal efficiency of 85% after 120 min of reaction [7]. Theoretically, the electrocatalytic oxidation (EO) technique is similar to the principle of photocatalysis. The organic degradation mechanisms of EO can mainly be divided into two processes, namely, direct and indirect electrochemical oxidation [8–11]: (i) direct electrochemical oxidation is the electron transfer directly to the anode after pollutant being adsorbed on the anode surface, without the involvement of other substances; (ii) indirect electrochemical oxidation occurs via the production of hydroxyl radicals ($\bullet OH$) which are produced by the discharge of water at the anode surface ($MO_x$), and subsequently attack organic pollutants. Moreover, according to the formation of different heterogeneous species, two approaches have been proposed for organic wastewater treatment by EO, namely, electrochemical conversion and electrochemical combustion. The electrochemical conversion is adsorbed hydroxyl radicals strongly reacting with the electrode to form higher oxide or superoxide ($MO_{x+1}$). The $MO_{x+1}/MO_x$ species on the surface is called chemisorbed active oxygen, which can convert or selectively oxidize organic pollutants. The other mechanism, electrochemical combustion, is the non-selective oxidation of pollutants through physical adsorbed active oxygen (physisorbed $MO_x(\bullet OH)$), resulting in being entirely mineralized to carbon dioxide, water, and inorganic ions. The above two approaches can partially or entirely oxidize organic matter. The $\bullet OH$ is the crucial substance of rapid degradation of organics. According to our previous studies, the yield of hydroxyl radicals produced by electrocatalysis is higher than that of photocatalysis ($4.5 \times 10^{-5}$ M/W cm$^2$ and $3.9 \times 10^{-5}$ M/W cm$^2$) under the same applied energy and catalyst surface area [12]. Thus, these results show that electrocatalysis has more significant potential to degrade pollutants. It has been proven to be effective in destroying various organic pollutants [13–16]. Furthermore, the EO has the advantage of easy operation, simple equipment, and less sludge production [17,18].

Although the EO to decompose the organic substances is promising, some application limits should be considered and overcome comprise: (1) the catalyst electrode is highly active and can be used for a long time, which can degrade pollutants stably and continuously; (2) appropriate electrical voltage enhances the pollutant degradation efficiency; (3) the influence of electrolyte on the conductivity of the solution and the transfer rate of

ion migration; (4) the feasibility of the EO method: treatment efficiency and operating cost. As a result, the fast and effective degradation of organic substance wastewater is difficult to be achieved by the not properly operated the EO system. That is, the catalyst electrode may be depleted due to long-term use and the complex properties of wastewater (such as extreme pH and corrosive conditions) in the electrocatalysis operation process. In addition, the decomposition of water increased with the introduction of intense electric voltage, but the removal rate of organic pollutants may not increase proportionally. At the same time, the water decomposition will change the pH of wastewater, resulting in difficult treatment and increased operational costs [19]. The EO technique was applied to treat municipal sewage, dye wastewater, and printing wastewater in our previous work. According to our results, the organic pollutants could be degraded effectively, and the operating conditions and removal mechanisms were studied. Therefore, it should be highly expected to apply the EO technique to treat wastewater containing DBP based on our past degradation experience.

For rapid organic contaminated degradation, one of the critical factors is to accelerate the removal rate of pollutants in the surface activity of catalyst electrode during electrocatalysis process. Accordingly, many materials such as $TiO_2$, $RuO_2$, $IrO_2$, and boron-doped diamond electrodes (BDD) have been extensively studied [20,21]. In recent years, composite material electrodes have shown more enhanced stability and activity than sole material and have become inevitable substitutes, such as $RuO_2$-$SnO_2$, $RuO_2$@$TiO_2$, $IrO_2$-$Ta_2O_5$, $RuO_2$-$IrO_2$-$TiO_2$ [22–25]. Among them, the $IrO_2$ with acid/alkali-resistance, corrosion resistance, and high chemical stability [26], which is higher than that of the $RuO_2$ electrode [27], especially shows better performance after being combined with $Ta_2O_5$ [24]. The oxide electrode of iridium combined with tantalum has been proven to possess a high activity [28]. In terms of using $IrO_2$ and $Ta_2O_5$ in the electrochemical, there have been studies to test the $IrO_2$-$Ta_2O_5$/Ti as the anode to remove organic pollutants from wastewater, such as the application of polluted river water, phenazopyridine hydrochloride (PhP), and dairy wastewater [29–31]. Moreover, from the literature, we obtained that the current change of the $IrO_2$-$Ta_2O_5$/Ti electrode was small when the voltage was high, showing its good stability. The high oxygen evolution potential can reduce the occurrence of oxygen evolution side reaction, which increases the generation of hydroxyl radicals. It is known from the voltammetry curves and Tafel curves that the $IrO_2$-$Ta_2O_5$/Ti electrode exhibits good electrochemical performance. In addition, the curve of $IrO_2$-$Ta_2O_5$/Ti contains a larger area at a higher scanning speed, indicating that the internal resistance of the electrode is small, and the symmetry is favorable. It indicated that the $IrO_2$-$Ta_2O_5$/Ti anode had good reversibility [29]. However, it is by far, and there have been no reports on using this electrode to electrocatalysis the DBP pollutants. The novelty of this research lies in establishing a practical and potential treatment technique for removing the refractory DBP. Through the integration of the high activity $IrO_2$-$Ta_2O_5$/Ti electrode and electrocatalyst system, the removal rate of DBP can be effectively improved, and the operation time is significantly shortened, which becomes a cost-effective processing technique. In this work, the variation of water quality, DBP mineralization, and removal efficiency of DBP concentration were investigated. Moreover, the stability of the $IrO_2$-$Ta_2O_5$/Ti electrode against DBP degradation was also confirmed through repeated experiments.

## 2. Results and Discussion

### 2.1. Identification of $IrO_2$-$Ta_2O_5$/Ti Electrode

First, the commercial $IrO_2$-$Ta_2O_5$/Ti electrode used in this study was confirmed by EDS to contain C, O, Ti, Ta, and Ir elements, as shown in Figure 1a, in which the weight percentages of each component were 3.57%, 21.86%, 0.33%, 47.21%, and 27.53%. Moreover, the XRD spectrum of the $IrO_2$-$Ta_2O_5$/Ti electrode is presented in Figure 1b. Three peaks centered at 41°, 55°, and 58° in the 2-theta belonged to polycrystalline $IrO_2$, while peaks of 63°, 71°, and 78° in the 2-theta were shown for the substrate titanium [32–34]. However, there is no notable peak observed for $Ta_2O_5$; the main reason for this finding can be

speculated as being the amorphous nature of $Ta_2O_5$ in the coating and the formation of $IrO_2$-$Ta_2O_5$ [35]. As a consequence, the presence of tantalum in the electrode was identified by EDS. In addition, since the coating content of the $IrO_2$ is 40 g/m$^2$ in the electrode surface and the density is 11.66 g/cm$^3$, the coating thickness of the electrode is about 3.6 μm after preliminary calculation.

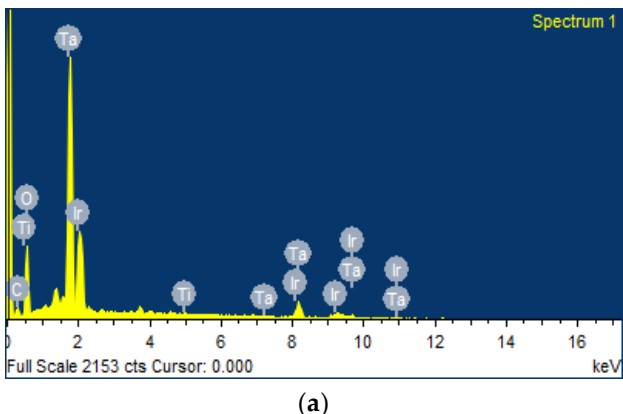

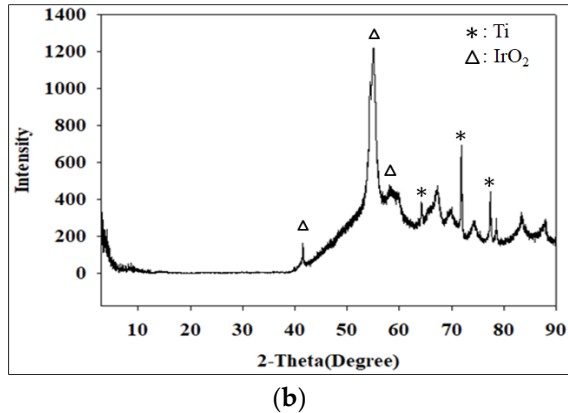

(**a**)  (**b**)

**Figure 1.** The element and microstructural analysis of $IrO_2$-$Ta_2O_5$/Ti electrode: (**a**) EDS spectrum; (**b**) XRD diffraction spectrum.

### 2.2. The influence of Electrode Material and Other Factors

Apart from the electrocatalytic oxidation reaction, the degradation of DBP may also be attributed to environmental factors such as natural photolysis and volatilize. In view of this, at the beginning of the experiment, this study has measured the effect of lab fluorescent light sources on the degradation of DBP under the presence of a $IrO_2$-$Ta_2O_5$/Ti electrode without electrocatalysis (without applied current). Meanwhile, the wastewater was heated (temperature increased from 20 to 80 °C) without electrocatalysis to observe the volatilization phenomena of DBP concentration with temperature, as shown in Figure 2. From the blank curve, the concentration of DBP is not changed during 60 min without electrocatalytic operation, which indicates that the effect of fluorescent light on the subsequent degradation of DBP can be ignored. Likewise, the concentration of DBP at the increased temperature range of 20~80 °C was changed insignificantly, which expresses that natural volatilization of DBP was not observed in the different temperatures. Similar results were also found in the study of $TiO_2$ photocatalysis [36]. From the results could be confirmed that the removal efficiency of DBP was not affected by natural photolysis and volatilize.

Further, observe the difference of catalyst material ($IrO_2$-$Ta_2O_5$/Ti electrode) and non-catalyst material (stainless steel electrode) on DBP degradation. Figure 2 shows that the DBP concentration decreased from 10 mg/L to 5.42 mg/L by stainless steel after 60 min of operation, and the removal efficiency was 46%. The degradation of DBP by stainless steel was mainly contributed by the direct anodic oxidation of the electrode surface. In contrast, using the Ir-Ta/Ti electrode can significantly improve the removal efficiency. DBP concentration decreased from 10 mg/L to 1.02 mg/L, and the removal efficiency reached 90%. Since the $IrO_2$-$Ta_2O_5$/Ti electrode belongs to the catalyst material, it mainly uses the generated hydroxyl radicals to oxidize pollutants indirectly. In this study, the main component of the $IrO_2$-$Ta_2O_5$/Ti electrode $IrO_2$ possesses conductive and active properties in the coating surface, which plays a role in conduction and electrocatalysis, and is the main contribution to the oxidation degradation of DBP; the other composition, $Ta_2O_5$, is non-conductive and has inert properties, high chemical stability, its main function being to stabilize the $IrO_2$ coating. Besides, the $Ta_2O_5$ can improve the dispersion of $IrO_2$ grains, which change the electrode morphology and the electrochemically active surface area can be increased [33].

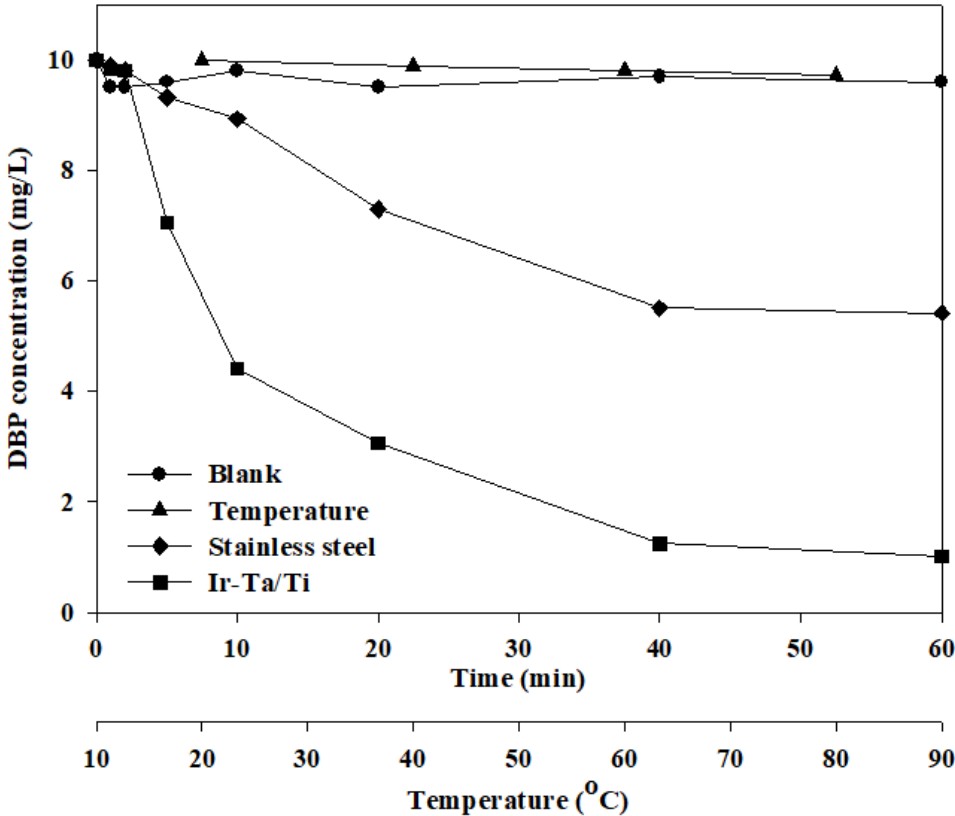

**Figure 2.** The removal efficiency of different electrodes for DBP degradation.

According to previous studies, under a voltage gradient of 11 V/cm, the yield of hydroxyl radicals produced by the $IrO_2$ electrode was significantly higher than that of the graphite electrode (non-catalyst), $5.24 \times 10^{-3}$ M and $0.86 \times 10^{-3}$ M, respectively [12]. Because of this, the $IrO_2$-$Ta_2O_5$/Ti electrode has a removal efficiency of 44% higher than the stainless steel electrode, which is mainly presumed to be the result of the increase in the yield of hydroxyl radicals by the $IrO_2$-$Ta_2O_5$/Ti electrode. Moreover, the higher DBP removal efficiency of the $IrO_2$-$Ta_2O_5$/Ti electrode can indirectly prove that organic pollutants are degraded by OH radicals in addition to the direct oxidation reaction on the electrode surface. As for the relationship between the reaction rate and temperature in the electrocatalytic operation process, it can be explained by being calculated through the Arrhenius equation. Based on current treatment results, the removal efficiency of DBP pollutants reached 90% within 60 min of EO operation, and the residual concentration (1.02 mg/L) has closed to the discharge standard of Taiwan's effluent (0.4 mg/L).

### 2.3. The Influence of Electrolyte and Voltage Gradient

Figure 3 shows the variation of DBP concentration with different electrolyte concentrations under a voltage gradient of 10 V/cm. It can be seen that the concentration of DBP decreased from the initial 10 mg/L to 5.35 mg/L under 0.001 M $Na_2SO_4$; the removal efficiency was 46.5%. Compared to 0.001 M of electrolyte concentration, 0.005 M $Na_2SO_4$ has a removal efficiency of 90%, so that DBP concentration decreased from 10 mg/L to 1.02 mg/L. The removal efficiency of DBP was increased with electrolyte concentration, from 46.5% of 0.001 M to 90% of 0.005 M. It is mainly due to the higher conductivity of 0.005 M leading to the higher current under a fixed voltage gradient and increasing the movement rate of ions, which was favorable to accelerate for DBP pollutants contact with the electrode surface, thus increasing the degradation efficiency of DBP. However, continuously increased electrolyte concentration has no positive effect on the removal of DBP. The removal efficiency of 0.01 M and 0.05 M $Na_2SO_4$ was 65% and 47%, respectively.

The main reason might be that high electrolyte concentration caused water electrolysis to be violent, which much oxygen bubble produced on the anode, resulting in the DBP being difficult to close the anode surface, thus decreasing the degradation efficiency of DBP. As a consequence, 0.005 M was the suitable electrolyte concentration for this study. As for the change of pH with time, there is an insignificant difference in the variation trend of different electrolyte concentrations. After 60 min, the pH was between 4.0 and 4.5; the rising conductivity trend is similar under different electrolyte concentrations (data not shown).

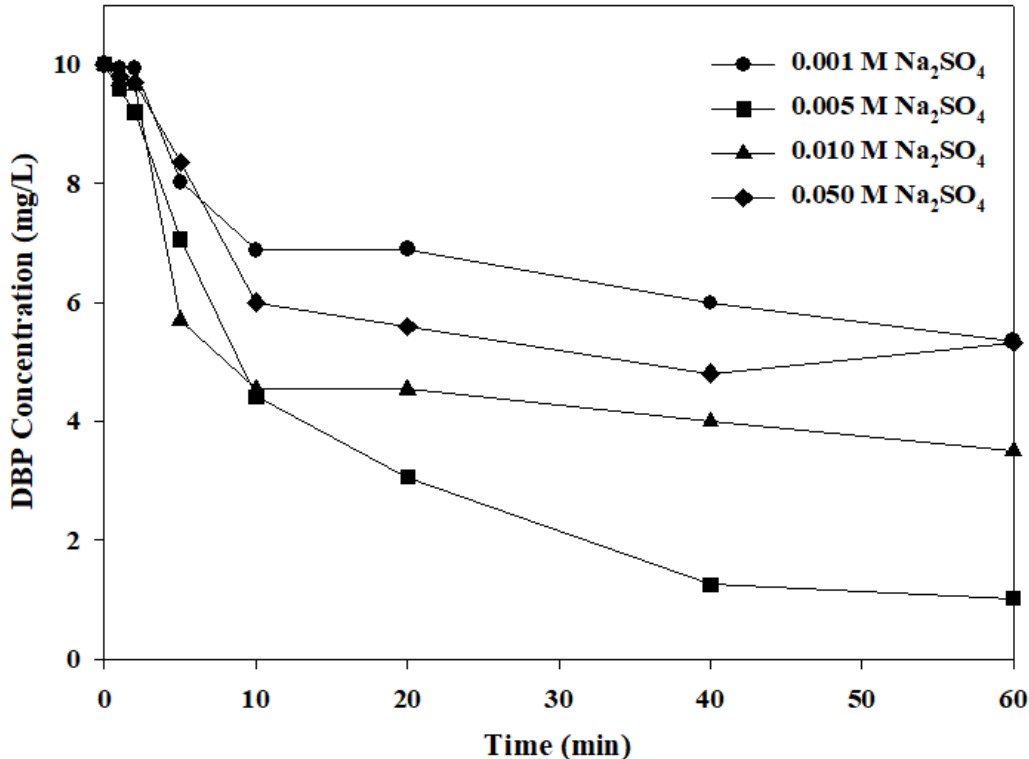

**Figure 3.** The variation of DBP concentration with different electrolyte concentrations.

Figure 4 shows the variation of pH and conductivity with different voltage gradients under a concentration of 0.005 M $Na_2SO_4$. From Figure 4, it can be observed that the voltage gradients of 8 V/cm, 9 V/cm, and 10 V/cm dropped from the initial pH 5.51 to 4.2, 4.28, and 4.5. The wastewater pH shows a slightly decreased trend under different voltage gradients, and increasing the voltage gradient has an insignificant effect on the pH. The pH decrease should be the result of the hydrogen ions generated at the anode end. However, the pH of wastewater is neutralized through the generation of hydroxyl radicals and the hydroxide ions, leading to the subsequent pH changes gradually gentle. It is observed that the pH of a voltage gradient of 10 V/cm decreases less obviously, which is presumed to be the result of the higher yield of hydroxyl radicals under the high voltage gradient. In addition, it can also be seen from Figure 4 that conductivity has slightly varied from 1.093 mS/cm to 1.123 mS/cm in 60 min under the voltage gradient of 10 V/cm. The conductivity shows a steady trend under different voltage gradients. The flatness of the conductivity was mainly attributed to stable pH in the wastewater. From the changes in pH and conductivity, the water quality parameters of the $IrO_2$-$Ta_2O_5$/Ti electrocatalytic system were relatively stable.

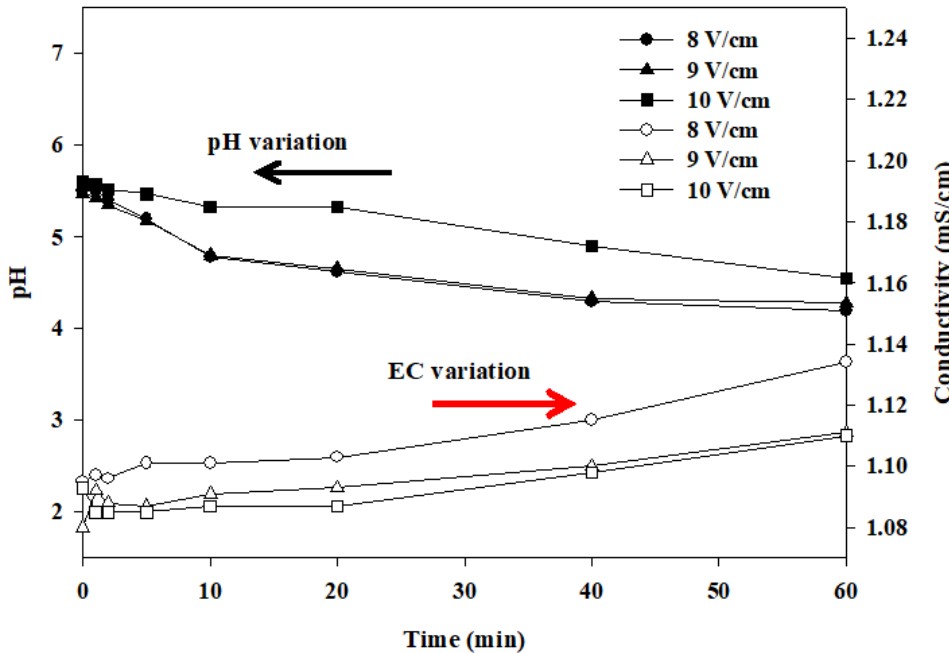

**Figure 4.** The tread of pH and conductivity with different voltage gradients.

From Figure 5, it is known that the electrocatalytic oxidation technique can effectively degrade DBP under different voltage gradients. For 60 min, the concentration of DBP from 10 mg/L reduced to 2.47 mg/L under 8 V/cm of voltage gradients, and the removal efficiency was 75.3%; the concentration of DBP decreased to 2.05 mg/L under 9 V/cm of voltage gradients, and the removal efficiency was 79.5%; the concentration of DBP reduced to 1.02 mg/L under 10 V/cm of voltage gradients, and the removal efficiency was 90%. The literature indicated that the •OH concentration can significantly be increased from $3.23 \times 10^{-3}$ M to $6.46 \times 10^{-3}$ M at 7 V/cm and 11 V/cm of voltage gradients [12]. As a result, the degradation of DBP was increased with the voltage gradient mainly due to applying the more significant energy in the electrocatalysis system, which produces high-yield hydroxyl radicals, thereby enhancing the degradation efficiency of DBP. By substituting degradation efficiency into the linear equation, there is a clear linear trend; $R^2 = 0.964$ could be obtained, showing that as the voltage gradient is increased, the DBP removing efficiency increases. This trend indicates that the voltage gradient strongly impacts the electrocatalytic system proportionally. Since the reaction of water electrolysis was increased significantly by continuously enhancing voltage gradients in the electrocatalysis system. The phenomenon might cause the degradation of DBP to be decreased, and the temperature was increased, which caused the higher electrode depletion and electric cost. To establish a practical electrocatalytic system stable and efficient for degrading the DBP, the 10 V/cm of the voltage gradient was a suitable range.

The degraded results are compared with the past report, as shown in Table 1. From the results described in Table 1, it is noted that the EO is sufficient to degrade water containing DBP because this EO mechanism almost wholly eliminates the pollutant concentration compared to other reports. Moreover, the energy consumption of the $IrO_2$-$Ta_2O_5$/Ti electrode under such operational conditions has been calculated. Based on the 90% DBP removal, the applied voltage, average current, and treatment time were 70 V, 1.0 A, and 40 min, respectively. After calculation, the energy consumption of the $IrO_2$-$Ta_2O_5$/Ti electrode was 0.047 KWh. Compared to the photocatalytic technique, electrocatalytic oxidation can save more than five times the electrical energy than photocatalysis (0.225~0.99 KWh) under similar removal efficiency [36–38]. The main reason for the shorter operation time and lower energy consumption of the EO is that the yield of hydroxyl radicals produced by electrocatalysis is higher than photocatalysis reactions. In addition, the direct oxidation

mechanism of the EO on the anode surface can also assist the degradation of organic matter. Therefore, it is known from the results that the EO technique with the $IrO_2$-$Ta_2O_5$/Ti electrode presents a cost-effective potential for DBP degradation.

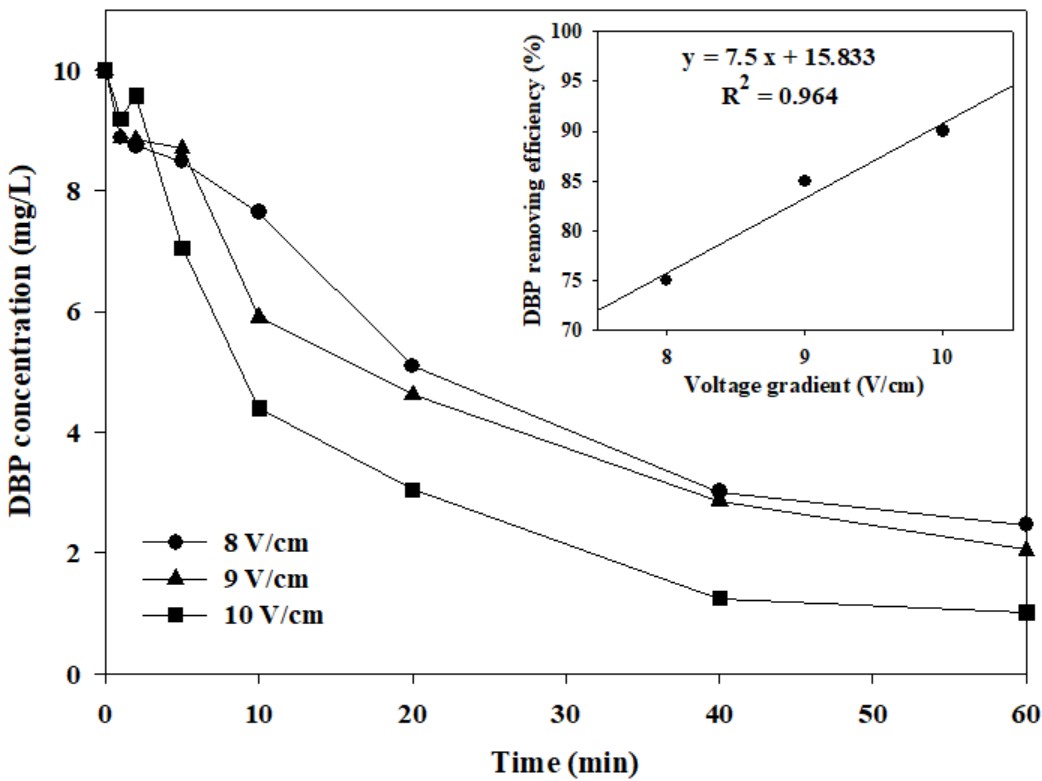

**Figure 5.** The variation of DBP concentration with different voltage gradients.

**Table 1.** The comparison table for the DBP treatment with previous reports.

| Treatment Method | Optimum Initial pH | Treatment Time (h) | Required Energy Consumption | % Pollutant Removal | Reference |
|---|---|---|---|---|---|
| Physical absorption | 13 | 2.0 | - | 97 | Wang and Chen (2015) [4] |
| Biological cultivation | 8 | 190 | - | 80 | Wu et al. (2013) [5] |
| Fe(III) Photocatalysis | 3 | 2 | Mercury lamp 125 W | 85 | Bajt et al. (2001) [37] |
| $TiO_2$ Photocatalysis | 6 | 1.0 | Xenon lamp 990 W | 90 | Kaneco et al. (2006) [36] |
| graphene/$TiO_2$ nanotube Photocatalysis | 11.5 | 1.5 | Xenon lamp 150 W | 95 | Wang et al. (2019) [38] |
| $IrO_2$-$Ta_2O_5$/Ti electrocatalysis | - | 0.67 | 70 W | 90 | Present work |

In addition to the degradation of DBP concentration, the variation of total organic carbon was also observed to understand the mineralization efficiency of DBP. The variation of TOC in different voltage gradients has been verified at a concentration of 0.005 M $Na_2SO_4$. It can be seen from Figure 6 that the TOC concentration shows a decreased trend under different voltage gradients. In 60 min, 8 V/cm, 9 V/cm, and 10 V/cm decreased from the initial 60 mg/L to 37 mg/L, 35 mg/L, and 27 mg/L, respectively, and the removal

efficiency was 38%, 47%, and 55%. This result indicated that the removal efficiency of TOC was increased with voltage gradient, which accelerates the mineralization efficiency of DBP. Meanwhile, the highly linear trend, $R^2$ = 0.999, could be obtained, showing that as the voltage gradient is increased, the TOC removing efficiency significantly increases. The electrophilic •OH played a vital role in the electrocatalytic oxidation degradation of DBP. The literature indicated that the degradation of DBP is by hydroxylation, oxidation, and dealkylation, and then undergoes bond breaking and opening ring to form the final mineralized products of $CO_2$ and $H_2O$ [38]. In the beginning, the highly oxidizing capacity hydroxyl radicals are generated on the electrode surface, which will attack the aliphatic chain and aromatic ring in the DBP structure and form bis(2-methoxyethyl) phthalate, 1,4-Dipropionyloxybenzene, 1-[3-Methoxy-4-(propionyloxy)phenyl]propyl propionate, and other intermediate products. These products will be converted into phthalic acid, butanoyl butanoate, and tetra-hydroxylated DBP, and then converted into formic acid, acetic acid, and acetaldehyde. Finally, the carboxylic derivatives form $CO_2$ and $H_2O$. According to our current degradation trend after 60 min of operation, it is known that extending the operation time to 120 min can continuously increase the degradation of the intermediate and achieve the complete mineralization of the DBP.

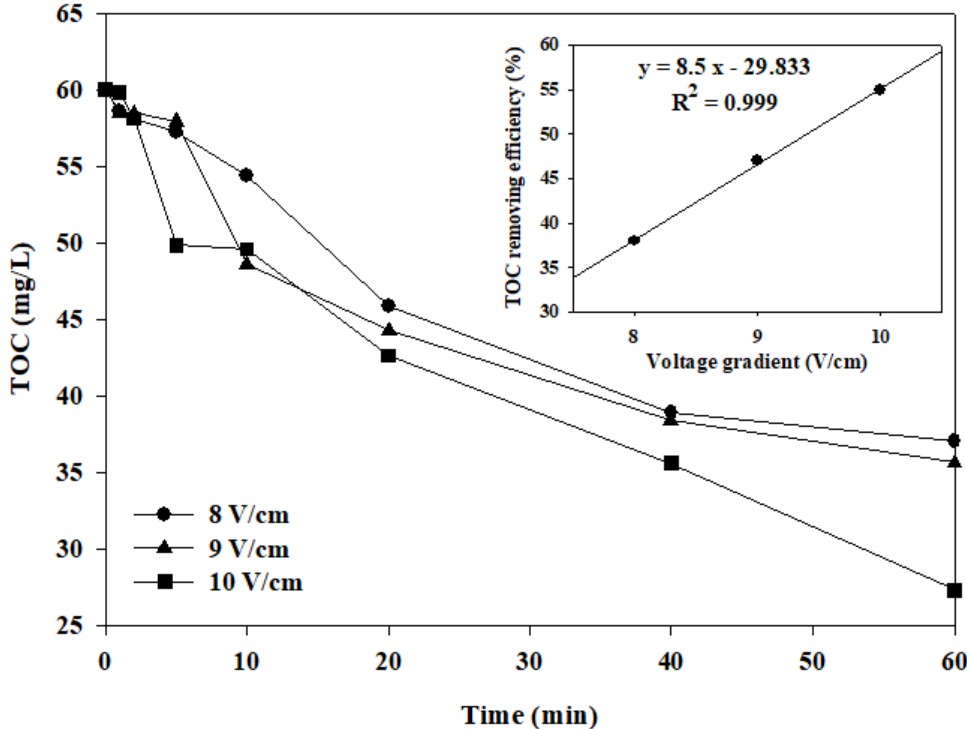

**Figure 6.** The variation of TOC concentration with different voltage gradients.

The mineralization current efficiency (MCE) of the DBP was calculated according to the following equation [11].

$$MCE\ (\%) = \frac{nFVs\ (\Delta TOC)}{4.32 * 10^7\,mIt} * 100 \tag{1}$$

where n is the number of electrons exchanged in the mineralization process of the DBP, F is the Faraday constant (96,487 C/mol), Vs is the solution volume (L), $\Delta TOC$ is TOC decays (mg/L), $4.32 \times 10^7$ is a conversion factor for units homogenization (= 3600 s/h × 12,000 mg C/mol), m is the number of carbon atoms of the DBP, and I is the applied current (A).

The number of exchanged electrons for the mineralization of an organic compound can be calculated from the following electrochemical reaction [10].

$$CxHyOz + (2x − z) H_2O + x CO_2 + (4x + y − 2z) H^+ + (4x + y − 2z) e^- \qquad (2)$$

since the molecular formula of DBP pollutant is $C_{12}H_{22}O_4$, and the number of exchanged electrons is 78; by substituting the relevant values into equation 1, where n = 78, ΔTOC is 33 mg/L, Vs = 0.5 L, I = 1, t = 1. After calculation, the mineralization current efficiency of DBP in this study is about 20%.

### 2.4. Surface Micrograph and Repeated Tests

To further confirm the electrocatalytic stability of the $IrO_2$-$Ta_2O_5$/Ti electrode to degrade the DBP, it has been tested, repeating three times under the conditions of 0.005 M and 10 V/cm. According to the results, the removal efficiency of the DBP can still reach about 90% under the repeated electrocatalysis (data not shown), which confirms the high activity performance of the $IrO_2$-$Ta_2O_5$/Ti electrode. Moreover, the $IrO_2$-$Ta_2O_5$/Ti electrode was a continuous operation for 100 h to observe the surface morphology of the electrode. According to the FESEM results shown in Figure 7, the surface was clean before the process, with only tiny traces and raised particles caused by the coating (Figure 7A). The surface morphology is composed of cracks and flat areas on the whole. The micro fissures or cracks structure in the electrode surface, which could enhance the electrode surface area and electrocatalytic activity for oxygen evolution [32]. After 100 h of operation (Figure 7B), there is no noticeable change on the surface except some dirt on the electrode, showing the stable structure of the $IrO_2$-$Ta_2O_5$/Ti electrode. The $IrO_2$-$Ta_2O_5$/Ti electrode stable operation is probably due to the existence of suitable quantities of inert $Ta_2O_5$, which form a typical morphology of cracks and solid solution structure. Therefore, the solid solution structure improves the electrode corrosion resistance effectively [32], and was favorable for long-time electrocatalytic operation. Meanwhile, the weight percentage of the Ir and Ta elements on the electrode surface was lost within about 2% after operation, according to the EDS spectrum (as shown in Table 2). Based on the physical–chemical properties of the electrode surface indicated that $IrO_2$-$Ta_2O_5$/Ti electrode could degrade DBP pollutants stably and effectively.

**Table 2.** The percentage of elements before and after electrode operation.

| Element | Weight (%) before Operation | Weight (%) after Operation | Difference (%) |
|---------|-----------------------------|----------------------------|----------------|
| C | 3.57 | 4.58 | 1.01 |
| O | 21.36 | 22.80 | 1.44 |
| Ti | 0.33 | 0.65 | 0.32 |
| Ta | 47.21 | 45.31 | 1.9 |
| Ir | 27.53 | 26.66 | 0.87 |
| Total | 100 | 100 | 1.01 |

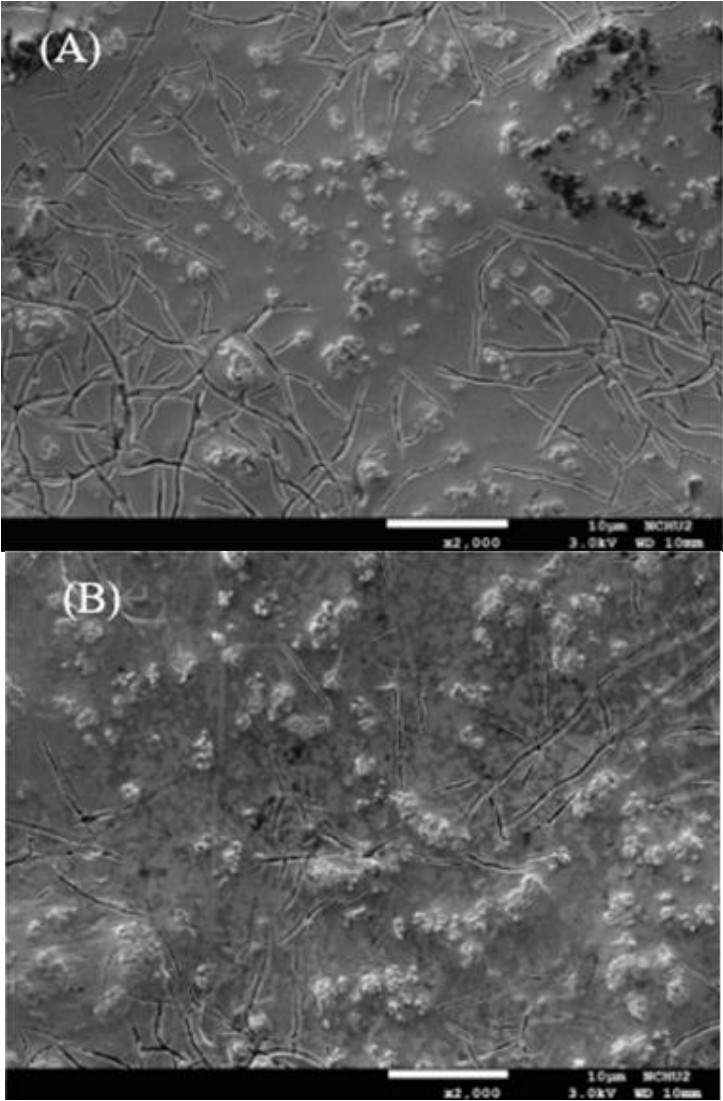

**Figure 7.** The micrograph of IrO$_2$-Ta$_2$O$_5$/Ti electrode (**A**) before operation; (**B**) after 100 h operation.

## 3. Materials and Method

### 3.1. Electrocatalytic Reactor

Dibutyl phthalate (DBP) was the target pollutant of the electrocatalytic process in this study. Table 3 lists the chemical structure and basic properties of DBP. The sodium sulfate (Na$_2$SO$_4$, purity 98%, Merck) was used as the electrolyte of the electrocatalytic experiment, and the DBP was extracted by ethyl acetate solvent (C$_4$H$_8$O$_2$, purity 99.5%, ECHO Chemical). In this work, a round glass reactor with 9 cm (diameter) × 18 cm (height) was utilized, and its effective volume was approximately 500 mL, as shown the Figure 8. The reactor comprised one pair of electrodes, in which commercial iridium–tantalum/titanium (IrO$_2$-Ta$_2$O$_5$/Ti) was employed as the anode manufactured by the sintering method. At the same time, graphite served as the cathode for all experiments. The size of both electrodes was identical (10 cm L × 2.5 cm W and 0.2 cm D). In order to obtain a suitable electrocatalytic operation, the distance between the anode and cathode was 7.0 cm and referred to our previous studies [19,39]. A direct-current (DC) power supply (GR Instek, GPR-20H20D) was used to provide electricity for electrocatalytic electrodes. An electromagnetic stirrer (Corning, Stirrer/Hot) was used to reach uniform mixing of wastewaters.

**Table 3.** Basic characteristics of DBP.

| Structure | |
|---|---|
| Formula | $C_{16}H_{22}O_4$ |
| Molecular weight | 278.34 |
| Density | 1.043 g/cm$^3$ |
| Dye content | 85–95% |
| Solubility | 13 mg/L |
| CAS NO. | 84-74-2 |
| Source | Riedel-deHaen |

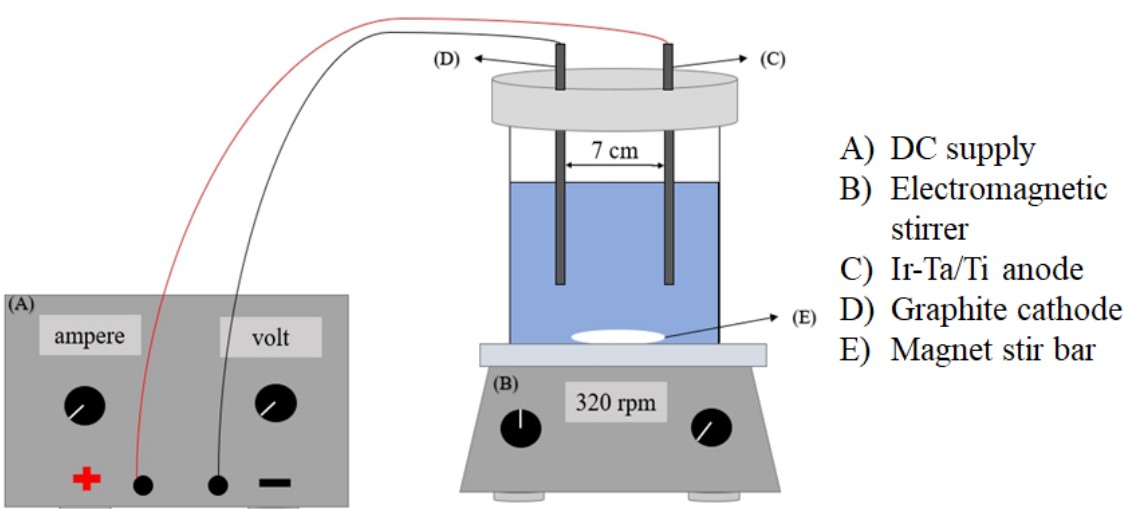

**Figure 8.** Schematic diagram of the electrocatalytic system.

*3.2. Electrocatalytic Degradation of DBP*

Table 4 shows the operating conditions of the electrocatalytic process. The first part of the experiment is to adjust the electrolyte concentration (0.001~0.05 M) and the voltage gradient (8~10 V/cm) to obtain the appropriate operating parameters for DBP degradation; the second part is to test the stability of the degrading DBP by an iridium-tantalum electrode. All degradation experiments were carried out in $Na_2SO_4$ electrolyte, and changes in DBP and TOC have been detected in fixed time intervals. The total operation time was 60 min. In this study, 10 mL samples were collected each time into a 50 mL brown glass bottle, and then 10 mL of ethyl acetate (volume ratio 1:1) was added. The DBP of water was extracted by mixing in the shaker at 150 rpm for 2 h; then, 2 mL of the supernatant was taken after standing for 1 hr, and DBP concentration was detected using a gas chromatography flame ion detector(GC-FID, Varian 3800 GC). The detection procedure of GC-FID: First, inject a sample of 3 μL DBP into the injector to evaporate the liquid, and use nitrogen as a carrier gas to bring it into the analysis column (DB-5 columns) at a fixed flow rate and temperature. Then, the temperature of the analysis column takes a heating gradient mode. The temperature was increased from initial 50 °C to the final temperature of 280 °C (rate of 12 °C/5 min), and after reaching 280 °C, held on for 5 min. Finally, the retention time of DBP was determined by the detector (Thermo TVA 1000B) and the concentration was quantified. The temperature of the injector and the detector was set to 280 °C. Furthermore, the mineralization of DBP was detected with a total organic carbon analyzer (TOC-VCSN,

Shimadzu, Japan). Finally, the removal efficiency of DBP concentration and the TOC were calculated using the following equation [40].

$$\text{Removal efficiency (\%)} = (C_o - C_t)/C_o \times 100 \qquad (3)$$

where $C_o$ and $C_t$ are the DBP and TOC concentration at the time of t = 0 and t, respectively.

**Table 4.** The operation conditions of electrocatalytic experiments.

| Item | Parameters |
|---|---|
| Electrode | Anode: $IrO_2$-$Ta_2O_5$/Ti Cathode: graphite |
| Voltage gradient (V/cm) | 8, 9, 10 |
| Electrolyte (M) | 0.001, 0.005, 0.01, 0.05 |
| DBP concentration (mg/L) | 10 |
| Treatment time (min) | 60 |

The pH, conductivity, and temperature were measured in each electrocatalytic experiment, apart from the DBP and TOC concentration analysis. Simultaneously, the electrical consumption was also calculated to estimate the economic competency of the $IrO_2$-$Ta_2O_5$/Ti electrode in treating the DBP. The formula for electric consumption is shown in the following equation [41], and the calculation result takes per kilowatt-hour (KWh) as the unit.

$$\text{Electrical consumption (KWh)} = UIT \qquad (4)$$

where U is the voltage (V), I is the average current (A), and T is the operating time (h).

## 4. Conclusions

After a series of experiments, the following conclusions were obtained:

1. The electrocatalytic oxidation (EO) technique treating water containing DBP, in which the pH and conductivity were slightly changed and eventually remained in a stable state.
2. $IrO_2$-$Ta_2O_5$/Ti electrode has the highest removal efficiency of DBP and can reach 90% under a voltage gradient of 10 V/cm and 0.005 M $Na_2SO_4$. The electric consumption has been calculated at about 0.046 KWh, and it is better than the current photocatalytic technique. It can provide a new approach for the degradation of DBP.
3. The removal efficiency of TOC can reach about 55% after the $IrO_2$-$Ta_2O_5$/Ti electrode for 60 min of treatment time. In addition, the mineralization reaction of DBP is increased with the voltage gradient.
4. The removal efficiency of the DBP can still be maintained at about 90% when the $IrO_2$-$Ta_2O_5$/Ti electrode is operated 3 times; the FESEM, EDS, and XRD patterns confirm that the surface structure of the $IrO_2$-$Ta_2O_5$/Ti electrode was still stable after use of 100 h.

**Author Contributions:** Conceptualization and methodology: Y.Z.; formal analysis, investigation, and data curation: J.-M.X. and S.-H.C.; validation: S.-Y.S.; writing—original draft preparation: S.-H.C. and S.-Y.S.; writing—review and editing: S.-Y.S., and M.K.; supervision: S.-Y.S. and M.K. All authors have read and agreed to the published version of the manuscript.

**Funding:** This research received no external funding.

**Conflicts of Interest:** The authors declare no conflict of interest.

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
