# Peer review of "Degradation of Dibutyl Phthalate Plasticizer in Water by High-Performance Iro2-Ta2O5/Ti Electrocatalytic Electrode"

_catalysts, doi:10.3390/catal11111368_

Round 1

Reviewer 1 Report

The manuscript reports the application of TrO2-TaO2O5/Ti electrodes to the treatment of DBP aqueous solution. however, it is not suitable for publication in Catalysts journal.

The discussion regarding the electrochemical oxidation of the DBP pollutant is poor. From the manuscript, it is hard to understand what is the novel contribution of this work (electrode was already used, the electrochemical approach was already used, electrochemical configuration reactor was already proposed by other authors, scale-up dimensions are not novel here). The authors must clearly indicate the novelty of their work. If the results of this manuscript are just a confirmation of the known results, then the manuscript is not original. For this reason, the manuscript should be revised, other comments are listed below:

1- Literature is not well cited since there are different articles published which discuss electrocatalytic oxidation, for example:

  • A. Martínez-Huitle, M. A. Rodrigo, I. Sires, O. Scialdone, Single and Coupled Electrochemical Processes and Reactors for the Abatement of Organic Water Pollutants: A Critical Review. Chem. Rev. 115 (2015) 13362-13407.
  • Brillas, C.A. Martínez-Huitle, Decontamination of wastewaters containing synthetic organic dyes by electrochemical methods. An updated review, Appl. Catal. B. 166-167 (2015) 603-643.
  • Panizza, G. Cerisola, Direct And Mediated Anodic Oxidation of Organic Pollutants. Chem. Rev. 109, 2009, 6541-6569.

2- Electrocatalytic fundamentals related to the environmental application should be discussed in the introduction

3- The given mechanism of the catalytic oxidation using such electrodes is wrong. The production of hydroxyl radicals is generated from the discharge of water:

3- The discussion is poor (Cyclic Voltammetry using such electrodes in the presence and the absence of the pollutant should be provided….)

4- Authors should estimate the Instantaneous Current Efficiency (ICE) from the COD data measured

Author Response

Dear Reviewer,

Please find the attachment.  Thank you.

Reviewer 2 Report

Authors present the study of the electrolytic decomposition of dibutyl phthalate (DBP) on the commercial IrO2-Ta2O5/Ti electrode. They study influence of the electrolyte concentration, pH and the voltage gradient on the removal of DBP. The results might be interesting from the technological point of view, but the data presentation is the weak point of this work. The manuscript is not suitable for publication in the current form. The points which should be improved are following:

  1. How the concentration of DBP was evaluated? The details should be given including the instrumentation.
  2. What is the meaning of colored areas in Figures 3, 4 and 5? Some plots (like 0.001M Na2SO4 in Fig. 3) are not visible at all.
  3. The thorough grammar check is needed. Some sentences are confusing, for example authors write: Meanwhile, the wastewater 157 was heated without electrocatalysis to simulate the temperature increase (20~80oC), 158 which observed the volatilization phenomena of DBP concentration with temperature in 159 the electrocatalytic oxidation process, as shown in Figure 2. What do you mean by heating to simulate the temperature increase? Unfortunately, there are more examples of oddly build sentences, therefore the whole text must be check carefully.
  4. The IrO2-Ta2O5/Ti electrode is commercial as stated in the text. Was is this electrode used for typically?

Author Response

(The authors gave the same response as above.)

Round 2

Reviewer 1 Report

After careful reading of the revised manuscript, the reviewer thinks that the paper can be accepted for publication.